# Plug-in Estimation in High-Dimensional Linear Inverse Problems: A Rigorous Analysis

**Alyson K. Fletcher**
Dept. Statistics
UC Los Angeles
akfletcher@ucla.edu

**Parthe Pandit**
Dept. ECE
UC Los Angeles
parthepandit@ucla.edu

**Sundeep Rangan**
Dept. ECE
NYU
srangan@nyu.edu

**Subrata Sarkar**
Dept. ECE
The Ohio State Univ.
sarkar.51@osu.edu

**Philip Schniter**
Dept. ECE
The Ohio State Univ.
schniter.1@osu.edu

## Abstract

Estimating a vector $\mathbf{x}$ from noisy linear measurements $\mathbf{Ax} + \mathbf{w}$ often requires use of prior knowledge or structural constraints on $\mathbf{x}$ for accurate reconstruction. Several recent works have considered combining linear least-squares estimation with a generic or "plug-in" denoiser function that can be designed in a modular manner based on the prior knowledge about $\mathbf{x}$. While these methods have shown excellent performance, it has been difficult to obtain rigorous performance guarantees. This work considers plug-in denoising combined with the recently-developed Vector Approximate Message Passing (VAMP) algorithm, which is itself derived via Expectation Propagation techniques. It shown that the mean squared error of this "plug-and-play" VAMP can be exactly predicted for high-dimensional right-rotationally invariant random $\mathbf{A}$ and Lipschitz denoisers. The method is demonstrated on applications in image recovery and parametric bilinear estimation.

## 1 Introduction

The estimation of an unknown vector $\mathbf{x}^0 \in \mathbb{R}^N$ from noisy linear measurements $\mathbf{y}$ of the form

$$\mathbf{y} = \mathbf{Ax}^0 + \mathbf{w} \in \mathbb{R}^M, \tag{1}$$

where $\mathbf{A} \in \mathbb{R}^{M \times N}$ is a known transform and $\mathbf{w}$ is disturbance, arises in a wide-range of learning and inverse problems. In many high-dimensional situations, such as when the measurements are fewer than the unknown parameters (i.e., $M \ll N$), it is essential to incorporate known structure on $\mathbf{x}^0$ in the estimation process. A fundamental challenge is how to perform structured estimation of $\mathbf{x}^0$ while maintaining computational efficiency and a tractable analysis.

*Approximate message passing* (AMP), originally proposed in [1], refers to a powerful class of algorithms that can be applied to reconstruction of $\mathbf{x}^0$ from (1) that can easily incorporate a wide class of statistical priors. In this work, we restrict our attention to $\mathbf{w} \sim \mathcal{N}(\mathbf{0}, \gamma_w^{-1}\mathbf{I})$, noting that AMP was extended to non-Gaussian measurements in [2, 3, 4]. AMP is computationally efficient, in that it generates a sequence of estimates $\{\widehat{\mathbf{x}}_k\}_{k=0}^{\infty}$ by iterating the steps

$$\widehat{\mathbf{x}}_k = \mathbf{g}(\mathbf{r}_k, \gamma_k) \tag{2a}$$

$$\mathbf{v}_k = \mathbf{y} - \mathbf{A}\widehat{\mathbf{x}}_k + \tfrac{N}{M}\langle \nabla \mathbf{g}(\mathbf{r}_{k-1}, \gamma_{k-1})\rangle \mathbf{v}_{k-1} \tag{2b}$$

$$\mathbf{r}_{k+1} = \widehat{\mathbf{x}}_k + \mathbf{A}^\mathsf{T}\mathbf{v}_k, \quad \gamma_{k+1} = M/\|\mathbf{v}_k\|^2, \tag{2c}$$

initialized with $\mathbf{r}_0 = \mathbf{A}^\mathsf{T}\mathbf{y}$, $\gamma_0 = M/\|\mathbf{y}\|^2$, $\mathbf{v}_{-1} = \mathbf{0}$, and assuming $\mathbf{A}$ is scaled so that $\|\mathbf{A}\|_F^2 \approx N$. In (2), $\mathbf{g} : \mathbb{R}^N \times \mathbb{R} \to \mathbb{R}^N$ is an estimation function chosen based on prior knowledge about $\mathbf{x}^0$, and $\langle \nabla \mathbf{g}(\mathbf{r}, \gamma) \rangle := \frac{1}{N} \sum_{n=1}^N \frac{\partial g_n(\mathbf{r}, \gamma)}{\partial r_n}$ denotes the divergence of $\mathbf{g}(\mathbf{r}, \gamma)$. For example, if $\mathbf{x}^0$ is known to be sparse, then it is common to choose $\mathbf{g}(\cdot)$ to be the componentwise soft-thresholding function, in which case AMP iteratively solves the LASSO [5] problem.

Importantly, for large, i.i.d., sub-Gaussian random matrices $\mathbf{A}$ and Lipschitz denoisers $\mathbf{g}(\cdot)$, the performance of AMP can be exactly predicted by a scalar *state evolution* (SE), which also provides testable conditions for optimality [6, 7, 8]. The initial work [6, 7] focused on the case where $\mathbf{g}(\cdot)$ is a separable function with identical components (i.e., $[\mathbf{g}(\mathbf{r}, \gamma)]_n = g(r_n, \gamma) \ \forall n$), while the later work [8] allowed non-separable $\mathbf{g}(\cdot)$. Interestingly, these SE analyses establish the fact that

$$\mathbf{r}_k = \mathbf{x}^0 + \mathcal{N}(\mathbf{0}, \mathbf{I}/\gamma_k), \tag{3}$$

leading to the important interpretation that $\mathbf{g}(\cdot)$ acts as a *denoiser*. This interpretation provides guidance on how to choose $\mathbf{g}(\cdot)$. For example, if $\mathbf{x}$ is i.i.d. with a known prior, then (3) suggests to choose a separable $\mathbf{g}(\cdot)$ composed of minimum mean-squared error (MMSE) scalar denoisers $g(r_n, \gamma) = \mathbb{E}(x_n | r_n = x_n + \mathcal{N}(0, 1/\gamma))$. In this case, [6, 7] established that, whenever the SE has a unique fixed point, the estimates $\widehat{\mathbf{x}}_k$ generated by AMP converge to the Bayes optimal estimate of $\mathbf{x}^0$ from $\mathbf{y}$. As another example, if $\mathbf{x}$ is a natural image, for which an analytical prior is lacking, then (3) suggests to choose $\mathbf{g}(\cdot)$ as a sophisticated image-denoising algorithm like BM3D [9] or DnCNN [10], as proposed in [11]. Many other examples of structured estimators $\mathbf{g}(\cdot)$ can be considered; we refer the reader to [8] and Section 5. Prior to [8], AMP SE results were established for special cases of $\mathbf{g}(\cdot)$ in [12, 13]. Plug-in denoisers have been combined in related algorithms [14, 15, 16].

An important limitation of AMP's SE is that it holds only for large, i.i.d., sub-Gaussian $\mathbf{A}$. AMP itself often fails to converge with small deviations from i.i.d. sub-Gaussian $\mathbf{A}$, such as when $\mathbf{A}$ is mildly ill-conditioned or non-zero-mean [4, 17, 18]. Recently, a robust alternative to AMP called *vector AMP* (VAMP) was proposed and analyzed in [19], based closely on expectation propagation [20]—see also [21, 22, 23]. There it was established that, if $\mathbf{A}$ is a large right-rotationally invariant random matrix and $\mathbf{g}(\cdot)$ is a separable Lipschitz denoiser, then VAMP's performance can be exactly predicted by a scalar SE, which also provides testable conditions for optimality. Importantly, VAMP applies to arbitrarily conditioned matrices $\mathbf{A}$, which is a significant benefit over AMP, since it is known that ill-conditioning is one of AMP's main failure mechanisms [4, 17, 18].

Unfortunately, the SE analyses of VAMP in [24] and its extension in [25] are limited to separable denoisers. This limitation prevents a full understanding of VAMP's behavior when used with non-separable denoisers, such as state-of-the-art image-denoising methods as recently suggested in [26]. The main contribution of this work is to show that the SE analysis of VAMP can be extended to a large class of non-separable denoisers that are Lipschitz continuous and satisfy a certain convergence property. The conditions are similar to those used in the analysis of AMP with non-separable denoisers in [8]. We show that there are several interesting non-separable denoisers that satisfy these conditions, including group-structured and convolutional neural network based denoisers.

An extended version with all proofs and other details are provided in [27].

## 2 Review of Vector AMP

The steps of VAMP algorithm of [19] are shown in Algorithm 1. Each iteration has two parts: A denoiser step and a Linear MMSE (LMMSE) step. These are characterized by *estimation functions* $\mathbf{g}_1(\cdot)$ and $\mathbf{g}_2(\cdot)$ producing estimates $\widehat{\mathbf{x}}_{1k}$ and $\widehat{\mathbf{x}}_{2k}$. The estimation functions take inputs $\mathbf{r}_{1k}$ and $\mathbf{r}_{2k}$ that we call *partial estimates*. The LMMSE estimation function is given by,

$$\mathbf{g}_2(\mathbf{r}_{2k}, \gamma_{2k}) := \left(\gamma_w \mathbf{A}^\mathsf{T}\mathbf{A} + \gamma_{2k}\mathbf{I}\right)^{-1} \left(\gamma_w \mathbf{A}^\mathsf{T}\mathbf{y} + \gamma_{2k}\mathbf{r}_{2k}\right), \tag{4}$$

where $\gamma_w > 0$ is a parameter representing an estimate of the precision (inverse variance) of the noise $\mathbf{w}$ in (1). The estimate $\widehat{\mathbf{x}}_{2k}$ is thus an MMSE estimator, treating the $\mathbf{x}$ as having a Gaussian prior with mean given by the partial estimate $\mathbf{r}_{2k}$. The estimation function $\mathbf{g}_1(\cdot)$ is called the *denoiser* and can be designed identically to the denoiser $\mathbf{g}(\cdot)$ in the AMP iterations (2). In particular, the denoiser is used to incorporate the structural or prior information on $\mathbf{x}$. As in AMP, in lines 5 and 11, $\langle \nabla \mathbf{g}_i \rangle$ denotes the normalized divergence.

**Algorithm 1** Vector AMP (LMMSE form)

---

**Require:** LMMSE estimator $\mathbf{g}_2(\cdot, \gamma_{2k})$ from (4), denoiser $\mathbf{g}_1(\cdot, \gamma_{1k})$, and number of iterations $K_{\mathrm{it}}$.
1: Select initial $\mathbf{r}_{10}$ and $\gamma_{10} \geq 0$.
2: **for** $k = 0, 1, \ldots, K_{\mathrm{it}}$ **do**
3:     // Denoising
4:     $\widehat{\mathbf{x}}_{1k} = \mathbf{g}_1(\mathbf{r}_{1k}, \gamma_{1k})$
5:     $\alpha_{1k} = \langle \nabla \mathbf{g}_1(\mathbf{r}_{1k}, \gamma_{1k}) \rangle$
6:     $\eta_{1k} = \gamma_{1k}/\alpha_{1k}, \gamma_{2k} = \eta_{1k} - \gamma_{1k}$
7:     $\mathbf{r}_{2k} = (\eta_{1k}\widehat{\mathbf{x}}_{1k} - \gamma_{1k}\mathbf{r}_{1k})/\gamma_{2k}$
8:
9:     // LMMSE estimation
10:     $\widehat{\mathbf{x}}_{2k} = \mathbf{g}_2(\mathbf{r}_{2k}, \gamma_{2k})$
11:     $\alpha_{2k} = \langle \nabla \mathbf{g}_2(\mathbf{r}_{2k}, \gamma_{2k}) \rangle$
12:     $\eta_{2k} = \gamma_{2k}/\alpha_{2k}, \gamma_{1,k+1} = \eta_{2k} - \gamma_{2k}$
13:     $\mathbf{r}_{1,k+1} = (\eta_{2k}\widehat{\mathbf{x}}_{2k} - \gamma_{2k}\mathbf{r}_{2k})/\gamma_{1,k+1}$
14: **end for**
15: Return $\widehat{\mathbf{x}}_{1K_{\mathrm{it}}}$.

---

The main result of [24] is that, under suitable conditions, VAMP admits a state evolution (SE) analysis that precisely describes the mean squared error (MSE) of the estimates $\widehat{\mathbf{x}}_{1k}$ and $\widehat{\mathbf{x}}_{2k}$ in a certain large system limit (LSL). Importantly, VAMP's SE analysis applies to arbitrary right rotationally invariant $\mathbf{A}$. This class is considerably larger than the set of sub-Gaussian i.i.d. matrices for which AMP applies. However, the SE analysis in [24] is restricted separable Lipschitz denoisers that can be described as follows: Let $g_{1n}(\mathbf{r}_1, \gamma_1)$ be the $n$-th component of the output of $\mathbf{g}_1(\mathbf{r}_1, \gamma_1)$. Then, it is assumed that,

$$\widehat{x}_{1n} = g_{1n}(\mathbf{r}_1, \gamma_1) = \phi(r_{1n}, \gamma_1), \tag{5}$$

for some function scalar-output function $\phi(\cdot)$ that does not depend on the component index $n$. Thus, the estimator is separable in the sense that the $n$-th component of the estimate, $\widehat{x}_{1n}$ depends only on the $n$-th component of the input $r_{1n}$ as well as the precision level $\gamma_1$. In addition, it is assumed that $\phi(r_1, \gamma_1)$ satisfies a certain Lipschitz condition. The separability assumption precludes the analysis of more general denoisers mentioned in the Introduction.

## 3 Extending the Analysis to Non-Separable Denoisers

The main contribution of the paper is to extend the state evolution analysis of VAMP to a class of denoisers that we call *uniformly Lipschitz* and *convergent under Gaussian noise*. This class is significantly larger than separable Lipschitz denoisers used in [24]. To state these conditions precisely, consider a sequence of estimation problems, indexed by a vector dimension $N$. For each $N$, suppose there is some "true" vector $\mathbf{u} = \mathbf{u}(N) \in \mathbb{R}^N$ that we wish to estimate from noisy measurements of the form, $\mathbf{r} = \mathbf{u} + \mathbf{z}$, where $\mathbf{z} \in \mathbb{R}^N$ is Gaussian noise. Let $\widehat{\mathbf{u}} = \mathbf{g}(\mathbf{r}, \gamma)$ be some estimator, parameterized by $\gamma$.

**Definition 1.** *The sequence of estimators* $\mathbf{g}(\cdot)$ *are said to be* *uniformly Lipschitz continuous* *if there exists constants A, B and C > 0, such that*

$$\|\mathbf{g}(\mathbf{r}_2, \gamma_2) - \mathbf{g}(\mathbf{r}_1, \gamma_1)\| \leq (A + B|\gamma_2 - \gamma_1|)\|\mathbf{r}_2 - \mathbf{r}_1\| + C\sqrt{N}|\gamma_2 - \gamma_1|, \tag{6}$$

*for any* $\mathbf{r}_1, \mathbf{r}_2, \gamma_1, \gamma_2$ *and* $N$.

**Definition 2.** *The sequence of random vectors* $\mathbf{u}$ *and estimators* $\mathbf{g}(\cdot)$ *are said to be convergent under Gaussian noise if the following condition holds: Let* $\mathbf{z}_1, \mathbf{z}_2 \in \mathbb{R}^N$ *be two se-* *quences where* $(z_{1n}, z_{2n})$ *are i.i.d. with* $(z_{1n}, z_{2n}) = \mathcal{N}(0, \mathbf{S})$ *for some positive definite covariance* $\mathbf{S} \in \mathbb{R}^{2 \times 2}$. *Then, all the following limits exist almost surely:*

$$\lim_{N \to \infty} \frac{1}{N}\mathbf{g}(\mathbf{u} + \mathbf{z}_1, \gamma_1)^{\mathsf{T}}\mathbf{g}(\mathbf{u} + \mathbf{z}_2, \gamma_2), \quad \lim_{N \to \infty} \frac{1}{N}\mathbf{g}(\mathbf{u} + \mathbf{z}_1, \gamma_1)^{\mathsf{T}}\mathbf{u}, \tag{7a}$$

$$\lim_{N \to \infty} \frac{1}{N}\mathbf{u}^{\mathsf{T}}\mathbf{z}_1, \quad \lim_{N \to \infty} \frac{1}{N}\|\mathbf{u}\|^2 \tag{7b}$$

$$\lim_{N \to \infty} \langle \nabla \mathbf{g}(\mathbf{u} + \mathbf{z}_1, \gamma_1) \rangle = \frac{1}{N S_{12}}\mathbf{g}(\mathbf{u} + \mathbf{z}_1, \gamma_1)^{\mathsf{T}}\mathbf{z}_2, \tag{7c}$$

*for all $\gamma_1, \gamma_2$ and covariance matrices $\mathbf{S}$. Moreover, the values of the limits are continuous in $\mathbf{S}$, $\gamma_1$ and $\gamma_2$.*

With these definitions, we make the following key assumption on the denoiser.

**Assumption 1.** *For each $N$, suppose that we have a "true" random vector $\mathbf{x}^0 \in \mathbb{R}^N$ and a denoiser $\mathbf{g}_1(\mathbf{r}_1, \gamma_1)$ acting on signals $\mathbf{r}_1 \in \mathbb{R}^N$. Following Definition 1, we assume the sequence of denoiser functions indexed by $N$, is uniformly Lipschitz continuous. In addition, the sequence of true vectors $\mathbf{x}^0$ and denoiser functions are convergent under Gaussian noise following Definition 2.*

The first part of Assumption 1 is relatively standard: Lipschitz and uniform Lipschitz continuity of the denoiser is assumed several AMP-type analyses including [6, 28, 24] What is new is the assumption in Definition 2. This assumption relates to the behavior of the denoiser $\mathbf{g}_1(\mathbf{r}_1, \gamma_1)$ in the case when the input is of the form, $\mathbf{r}_1 = \mathbf{x}^0 + \mathbf{z}$. That is, the input is the true signal with a Gaussian noise perturbation. In this setting, we will be requiring that certain correlations converge. Before continuing our analysis, we briefly show that separable denoisers as well as several interesting non-separable denoisers satisfy these conditions.

**Separable Denoisers.** We first show that the class of denoisers satisfying Assumption 1 includes the separable Lipschitz denoisers studied in most AMP analyses such as [6]. Specifically, suppose that the true vector $\mathbf{x}^0$ has i.i.d. components with bounded second moments and the denoiser $\mathbf{g}_1(\cdot)$ is separable in that it is of the form (5). Under a certain uniform Lipschitz condition, it is shown in the extended version of this paper [27] that the denoiser satisfies Assumption 1.

**Group-Based Denoisers.** As a first non-separable example, let us suppose that the vector $\mathbf{x}^0$ can be represented as an $L \times K$ matrix. Let $\mathbf{x}_\ell^0 \in \mathbb{R}^K$ denote the $\ell$-th row and assume that the rows are i.i.d. Each row can represent a *group*. Suppose that the denoiser $\mathbf{g}_1(\cdot)$ is *groupwise separable*. That is, if we denote by $\mathbf{g}_{1\ell}(\mathbf{r}, \ell)$ the $\ell$-th row of the output of the denoiser, we assume that

$$\mathbf{g}_{1\ell}(\mathbf{r}, \gamma) = \phi(\mathbf{r}_\ell, \gamma) \in \mathbb{R}^K, \tag{8}$$

for a vector-valued function $\phi(\cdot)$ that is the same for all rows. Thus, the $\ell$-th row output $\mathbf{g}_\ell(\cdot)$ depends only on the $\ell$-th row input. Such groupwise denoisers have been used in AMP and EP-type methods for group LASSO and other structured estimation problems [29, 30, 31]. Now, consider the limit where the group size $K$ is fixed, and the number of groups $L \to \infty$. Then, under suitable Lipschitz continuity conditions, the extended version of this paper [27] shows that groupwise separable denoiser also satisfies Assumption 1.

**Convolutional Denoisers.** As another non-separable denoiser, suppose that, for each $N$, $\mathbf{x}^0$ is an $N$ sample segment of a stationary, ergodic process with bounded second moments. Suppose that the denoiser is given by a linear convolution,

$$\mathbf{g}_1(\mathbf{r}_1) := T_N(\mathbf{h} * \mathbf{r}_1), \tag{9}$$

where $\mathbf{h}$ is a finite length filter and $T_N(\cdot)$ truncates the signal to its first $N$ samples. For simplicity, we assume there is no dependence on $\gamma_1$. Convolutional denoising arises in many standard linear estimation operations on wide sense stationary processes such as Weiner filtering and smoothing [32]. If we assume that $\mathbf{h}$ remains constant and $N \to \infty$, the extended version of this paper [27] shows that the sequence of random vectors $\mathbf{x}^0$ and convolutional denoisers $\mathbf{g}_1(\cdot)$ satisfies Assumption 1.

**Convolutional Neural Networks.** In recent years, there has been considerable interest in using trained deep convolutional neural networks for image denoising [33, 34]. As a simple model for such a denoiser, suppose that the denoiser is a composition of maps,

$$\mathbf{g}_1(\mathbf{r}_1) = (F_L \circ F_{L-1} \circ \cdots \circ F_1)(\mathbf{r}_1), \tag{10}$$

where $F_\ell(\cdot)$ is a sequence of layer maps where each layer is either a multi-channel convolutional operator or Lipschitz separable activation function, such as sigmoid or ReLU. Under mild assumptions on the maps, it is shown in the extended version of this paper [27] that the estimator sequence $\mathbf{g}_1(\cdot)$ can also satisfy Assumption 1.

**Singular-Value Thresholding (SVT) Denoiser.** Consider the estimation of a low-rank matrix $\mathbf{X}^0$ from linear measurements $\mathbf{y} = \mathcal{A}(\mathbf{X}^0)$, where $\mathcal{A}$ is some linear operator [35]. Writing the SVD of $\mathbf{R}$ as $\mathbf{R} = \sum_i \sigma_i \mathbf{u}_i \mathbf{v}_i^\mathsf{T}$, the SVT denoiser is defined as

$$\mathbf{g}_1(\mathbf{R}, \gamma) := \sum_i (\sigma_i - \gamma)_+ \mathbf{u}_i \mathbf{v}_i^\mathsf{T}, \tag{11}$$

where $(x)_+ := \max\{0, x\}$. In the extended version of this paper [27], we show that $\mathbf{g}_1(\cdot)$ satisfies Assumption 1.

# 4 Large System Limit Analysis

## 4.1 System Model

Our main theoretical contribution is to show that the SE analysis of VAMP in [19] can be extended to the non-separable case. We consider a sequence of problems indexed by the vector dimension $N$. For each $N$, we assume that there is a "true" random vector $\mathbf{x}^0 \in \mathbb{R}^N$ observed through measurements $\mathbf{y} \in \mathbb{R}^M$ of the form in (1) where $\mathbf{w} \sim \mathcal{N}(\mathbf{0}, \gamma_{w0}^{-1}\mathbf{I})$. We use $\gamma_{w0}$ to denote the "true" noise precision to distinguish this from the postulated precision, $\gamma_w$, used in the LMMSE estimator (4). Without loss of generality (see below), we assume that $M = N$. We assume that $\mathbf{A}$ has an SVD,

$$\mathbf{A} = \mathbf{U}\mathbf{S}\mathbf{V}^\mathsf{T}, \quad \mathbf{S} = \mathrm{diag}(\mathbf{s}), \quad \mathbf{s} = (s_1, \ldots, s_N), \tag{12}$$

where $\mathbf{U}$ and $\mathbf{V}$ are orthogonal and $\mathbf{S}$ is non-negative and diagonal. The matrix $\mathbf{U}$ is arbitrary, $\mathbf{s}$ is an i.i.d. random vector with components $s_i \in [0, s_{max}]$ almost surely. Importantly, we assume that $\mathbf{V}$ is Haar distributed, meaning that it is uniform on the $N \times N$ orthogonal matrices. This implies that $\mathbf{A}$ is *right rotationally invariant* meaning that $\mathbf{A} \overset{d}{=} \mathbf{A}\mathbf{V}_0$ for any orthogonal matrix $\mathbf{V}_0$. We also assume that $\mathbf{w}, \mathbf{x}^0, \mathbf{s}$ and $\mathbf{V}$ are all independent. As in [19], we can handle the case of rectangular $\mathbf{V}$ by zero padding $\mathbf{s}$.

These assumptions are similar to those in [19]. The key new assumption is Assumption 1. Given such a denoiser and postulated variance $\gamma_w$, we run the VAMP algorithm, Algorithm 1. We assume that the initial condition is given by,

$$\mathbf{r} = \mathbf{x}^0 + \mathcal{N}(\mathbf{0}, \tau_{10}\mathbf{I}), \tag{13}$$

for some initial error variance $\tau_{10}$. In addition, we assume

$$\lim_{N \to \infty} \gamma_{10} = \overline{\gamma}_{10}, \tag{14}$$

almost surely for some $\overline{\gamma}_{10} \geq 0$.

Analogous to [24], we define two key functions: *error functions* and *sensitivity functions*. The error functions characterize the MSEs of the denoiser and LMMSE estimator under AWGN measurements. For the denoiser $\mathbf{g}_1(\cdot, \gamma_1)$, we define the error function as

$$\mathcal{E}_1(\gamma_1, \tau_1) := \lim_{N \to \infty} \frac{1}{N}\|\mathbf{g}_1(\mathbf{x}^0 + \mathbf{z}, \gamma_1) - \mathbf{x}^0\|^2, \quad \mathbf{z} \sim \mathcal{N}(\mathbf{0}, \tau_1\mathbf{I}), \tag{15}$$

and, for the LMMSE estimator, as

$$\begin{aligned}
\mathcal{E}_2(\gamma_2, \tau_2) &:= \lim_{N \to \infty} \frac{1}{N}\mathbb{E}\|\mathbf{g}_2(\mathbf{r}_2, \gamma_2) - \mathbf{x}^0\|^2, \\
\mathbf{r}_2 &= \mathbf{x}^0 + \mathcal{N}(\mathbf{0}, \tau_2\mathbf{I}), \quad \mathbf{y} = \mathbf{A}\mathbf{x}^0 + \mathcal{N}(\mathbf{0}, \gamma_{w0}^{-1}\mathbf{I}).
\end{aligned} \tag{16}$$

The limit (15) exists almost surely due to the assumption of $\mathbf{g}_1(\cdot)$ being convergent under Gaussian noise. Although $\mathcal{E}_2(\gamma_2, \tau_2)$ implicitly depends on the precisions $\gamma_{w0}$ and $\gamma_w$, we omit this dependence to simplify the notation. We also define the *sensitivity functions* as

$$\mathcal{A}_i(\gamma_i, \tau_i) := \lim_{N \to \infty} \langle \nabla \mathbf{g}_i(\mathbf{x}^0 + \mathbf{z}_i, \gamma_i) \rangle, \quad \mathbf{z}_i \sim \mathcal{N}(\mathbf{0}, \tau_i\mathbf{I}). \tag{17}$$

## 4.2 State Evolution of VAMP

We now show that the VAMP algorithm with a non-separable denoiser follows the identical state evolution equations as the separable case given in [19]. Define the error vectors,

$$\mathbf{p}_k := \mathbf{r}_{1k} - \mathbf{x}^0, \quad \mathbf{q}_k := \mathbf{V}^{\mathsf{T}}(\mathbf{r}_{2k} - \mathbf{x}^0). \tag{18}$$

Thus, $\mathbf{p}_k$ represents the error between the partial estimate $\mathbf{r}_{1k}$ and the true vector $\mathbf{x}^0$. The error vector $\mathbf{q}_k$ represents the transformed error $\mathbf{r}_{2k} - \mathbf{x}^0$. The SE analysis will show that these errors are asymptotically Gaussian. In addition, the analysis will exactly predict the variance on the partial estimate errors (18) and estimate errors, $\widehat{\mathbf{x}}_i - \mathbf{x}^0$. These variances are computed recursively through what we will call the *state evolution* equations:

$$\overline{\alpha}_{1k} = \mathcal{A}_1(\overline{\gamma}_{1k}, \tau_{1k}), \quad \overline{\eta}_{1k} = \frac{\overline{\gamma}_{1k}}{\overline{\alpha}_{1k}}, \quad \overline{\gamma}_{2k} = \overline{\eta}_{1k} - \overline{\gamma}_{1k} \tag{19a}$$

$$\tau_{2k} = \frac{1}{(1 - \overline{\alpha}_{1k})^2} \left[ \mathcal{E}_1(\overline{\gamma}_{1k}, \tau_{1k}) - \overline{\alpha}_{1k}^2 \tau_{1k} \right], \tag{19b}$$

$$\overline{\alpha}_{2k} = \mathcal{A}_2(\overline{\gamma}_{2k}, \tau_{2k}), \quad \overline{\eta}_{2k} = \frac{\overline{\gamma}_{2k}}{\overline{\alpha}_{2k}}, \quad \overline{\gamma}_{1,k+1} = \overline{\eta}_{2k} - \overline{\gamma}_{2k} \tag{19c}$$

$$\tau_{1,k+1} = \frac{1}{(1 - \overline{\alpha}_{2k})^2} \left[ \mathcal{E}_2(\overline{\gamma}_{2k}, \tau_{2k}) - \overline{\alpha}_{2k}^2 \tau_{2k} \right], \tag{19d}$$

which are initialized with $k = 0$, $\tau_{10}$ in (13) and $\overline{\gamma}_{10}$ defined from the limit (14). The SE equations in (19) are identical to those in [19] with the new error and sensitivity functions for the non-separable denoisers. We can now state our main result, which is proven in the extended version of this paper [27].

**Theorem 1.** *Under the above assumptions and definitions, assume that the sequence of true random vectors $\mathbf{x}^0$ and denoisers $\mathbf{g}_1(\mathbf{r}_1, \gamma_1)$ satisfy Assumption 1. Assume additionally that, for all iterations $k$, the solution $\overline{\alpha}_{1k}$ from the SE equations (19) satisfies $\overline{\alpha}_{1k} \in (0, 1)$ and $\overline{\gamma}_{ik} > 0$. Then,*

*(a) For any $k$, the error vectors on the partial estimates, $\mathbf{p}_k$ and $\mathbf{q}_k$ in (18) can be written as,*

$$\mathbf{p}_k = \widetilde{\mathbf{p}}_k + O(\tfrac{1}{\sqrt{N}}), \quad \mathbf{q}_k = \widetilde{\mathbf{q}}_k + O(\tfrac{1}{\sqrt{N}}), \tag{20}$$

*where, $\widetilde{\mathbf{p}}_k$ and $\widetilde{\mathbf{q}}_k \in \mathbb{R}^N$ are each i.i.d. Gaussian random vectors with zero mean and per component variance $\tau_{1k}$ and $\tau_{2k}$, respectively.*

*(b) For any fixed iteration $k \geq 0$, and $i = 1, 2$, we have, almost surely*

$$\lim_{N \to \infty} \frac{1}{N} \|\widehat{\mathbf{x}}_i - \mathbf{x}^0\|^2 = \frac{1}{\overline{\eta}_{ik}}, \quad \lim_{N \to \infty} (\alpha_{ik}, \eta_{ik}, \gamma_{ik}) = (\overline{\alpha}_{ik}, \overline{\eta}_{ik}, \overline{\gamma}_{ik}). \tag{21}$$

In (20), we have used the notation, that when $\mathbf{u}, \widetilde{\mathbf{u}} \in \mathbb{R}^N$ are sequences of random vectors, $\mathbf{u} = \widetilde{\mathbf{u}} + O(\tfrac{1}{\sqrt{N}})$ means $\lim_{N \to \infty} \frac{1}{N} \|\mathbf{u} - \widetilde{\mathbf{u}}\|^2 = 0$ almost surely. Part (a) of Theorem 1 thus shows that the error vectors $\mathbf{p}_k$ and $\mathbf{q}_k$ in (18) are approximately i.i.d. Gaussian. The result is a natural extension to the main result on separable denoisers in [19]. Moreover, the variance on the variance on the errors, along with the mean squared error (MSE) of the estimates $\widehat{\mathbf{x}}_{ik}$ can be exactly predicted by the same SE equations as the separable case. The result thus provides an asymptotically exact analysis of VAMP extended to non-separable denoisers.

## 5 Numerical Experiments

### 5.1 Compressive Image Recovery

We first consider the problem of compressive image recovery, where the goal is to recover an image $\mathbf{x}^0 \in \mathbb{R}^N$ from measurements $\mathbf{y} \in \mathbb{R}^M$ of the form (1) with $M \ll N$. This problem arises in many imaging applications, such as magnetic resonance imaging, radar imaging, computed tomography, etc., although the details of $\mathbf{A}$ and $\mathbf{x}^0$ change in each case.

One of the most popular approaches to image recovery is to exploit sparsity in the wavelet transform coefficients $\mathbf{c} := \mathbf{\Psi}\mathbf{x}^0$, where $\mathbf{\Psi}$ is a suitable orthonormal wavelet transform. Rewriting (1) as

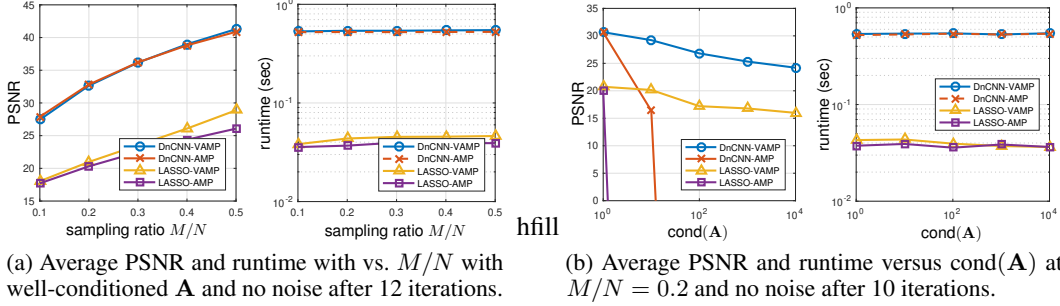

(a) Average PSNR and runtime with vs. $M/N$ with well-conditioned $\mathbf{A}$ and no noise after 12 iterations.

(b) Average PSNR and runtime versus cond($\mathbf{A}$) at $M/N = 0.2$ and no noise after 10 iterations.

Figure 1: Compressive image recovery: PSNR and runtime vs. rate $M/N$ and cond($\mathbf{A}$)

$\mathbf{y} = \mathbf{A}\mathbf{\Psi}\mathbf{c} + \mathbf{w}$, the idea is to first estimate $\mathbf{c}$ from $\mathbf{y}$ (e.g., using LASSO) and then form the image estimate via $\widehat{\mathbf{x}} = \mathbf{\Psi}^\mathsf{T}\widehat{\mathbf{c}}$. Although many algorithms exist to solve the LASSO problem, the AMP algorithms are among the fastest (see, e.g., [36, Fig.1]). As an alternative to the sparsity-based approach, it was recently suggested in [11] to recover $\mathbf{x}^0$ directly using AMP (2) by choosing the estimation function $\mathbf{g}$ as a sophisticated image-denoising algorithm like BM3D [9] or DnCNN [10].

Figure 1a compares the LASSO- and DnCNN-based versions of AMP and VAMP for $128{\times}128$ image recovery under well-conditioned $\mathbf{A}$ and no noise. Here, $\mathbf{A} = \mathbf{JPHD}$, where $\mathbf{D}$ is a diagonal matrix with random $\pm 1$ entries, $\mathbf{H}$ is a discrete Hadamard transform (DHT), $\mathbf{P}$ is a random permutation matrix, and $\mathbf{J}$ contains the first $M$ rows of $\mathbf{I}_N$. The results average over the well-known *lena*, *barbara*, *boat*, *house*, and *peppers* images using 10 random draws of $\mathbf{A}$ for each. The figure shows that AMP and VAMP have very similar runtimes and PSNRs when $\mathbf{A}$ is well-conditioned, and that the DnCNN approach is about 10 dB more accurate, but $10\times$ as slow, as the LASSO approach. Figure 2 shows the state-evolution prediction of VAMP's PSNR on the *barbara* image at $M/N = 0.5$, averaged over 50 draws of $\mathbf{A}$. The state-evolution accurately predicts the PSNR of VAMP.

To test the robustness to the condition number of $\mathbf{A}$, we repeated the experiment from Fig. 1a using $\mathbf{A} = \mathbf{J}\mathrm{Diag}(\mathbf{s})\mathbf{PHD}$, where $\mathrm{Diag}(\mathbf{s})$ is a diagonal matrix of singular values. The singular values were geometrically spaced, i.e., $s_m/s_{m-1} = \rho\ \forall m$, with $\rho$ chosen to achieve a desired cond($\mathbf{A}$) := $s_1/s_M$. The sampling rate was fixed at $M/N = 0.2$, and the measurements were noiseless, as before. The results, shown in Fig. 1b, show that AMP diverged when cond($\mathbf{A}$) $\geq 10$, while VAMP exhibited only a mild PSNR degradation due to ill-conditioned $\mathbf{A}$. The original images and example image recoveries are included in the extended version of this paper.

## 5.2 Bilinear Estimation via Lifting

We now use the structured linear estimation model (1) to tackle problems in *bilinear* estimation through a technique known as "lifting" [37, 38, 39, 40]. In doing so, we are motivated by applications like blind deconvolution [41], self-calibration [39], compressed sensing (CS) with matrix uncertainty [42], and joint channel-symbol estimation [43]. All cases yield measurements $\mathbf{y}$ of the form

$$\mathbf{y} = \big(\textstyle\sum_{l=1}^{L} b_l \mathbf{\Phi}_l\big)\mathbf{c} + \mathbf{w} \in \mathbb{R}^M, \tag{22}$$

where $\{\mathbf{\Phi}_l\}_{l=1}^L$ are known, $\mathbf{w} \sim \mathcal{N}(\mathbf{0}, \mathbf{I}/\gamma_w)$, and the objective is to recover both $\mathbf{b} := [b_1, \ldots, b_L]^\mathsf{T}$ and $\mathbf{c} \in \mathbb{R}^P$. This bilinear problem can be "lifted" into a linear problem of the form (1) by setting

$$\mathbf{A} = [\mathbf{\Phi}_1 \quad \mathbf{\Phi}_2 \quad \cdots \quad \mathbf{\Phi}_L] \in \mathbb{R}^{M \times LP} \text{ and } \mathbf{x} = \mathrm{vec}(\mathbf{c}\mathbf{b}^\mathsf{T}) \in \mathbb{R}^{LP}, \tag{23}$$

where $\mathrm{vec}(\mathbf{X})$ vectorizes $\mathbf{X}$ by concatenating its columns. When $\mathbf{b}$ and $\mathbf{c}$ are i.i.d. with known priors, the MMSE denoiser $\mathbf{g}(\mathbf{r}, \gamma) = \mathbb{E}(\mathbf{x}|\mathbf{r} = \mathbf{x} + \mathcal{N}(\mathbf{0}, \mathbf{I}/\gamma))$ can be implemented near-optimally by the rank-one AMP algorithm from [44] (see also [45, 46, 47]), with divergence estimated as in [11].

We first consider *CS with matrix uncertainty* [42], where $b_1$ is known. For these experiments, we generated the unknown $\{b_l\}_{l=2}^L$ as i.i.d. $\mathcal{N}(0, 1)$ and the unknown $\mathbf{c} \in \mathbb{R}^P$ as $K$-sparse with $\mathcal{N}(0, 1)$ nonzero entries. Fig. 2 shows that the MSE on $\mathbf{x}$ of lifted VAMP is very close to its SE prediction when $K = 12$. We then compared lifted VAMP to PBiGAMP from [48], which applies AMP directly to the (non-lifted) bilinear problem, and to WSS-TLS from [42], which uses non-convex optimization. We also compared to MMSE estimation of $\mathbf{b}$ under oracle knowledge of $\mathbf{c}$, and MMSE

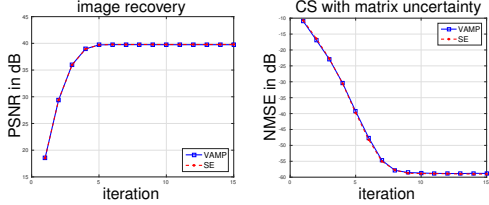

Figure 2: SE prediction & VAMP for image recovery and CS with matrix uncertainty

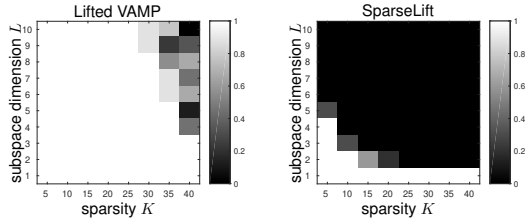

Figure 3: Self-calibration: Success rate vs. sparsity $K$ and subspace dimension $L$

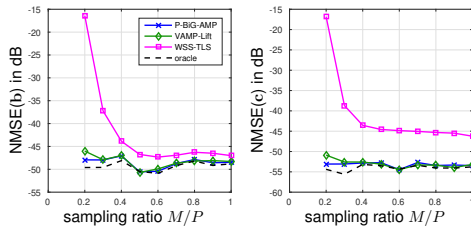

(a) NMSE vs. $M/P$ with i.i.d. $\mathcal{N}(0,1)$ $\mathbf{A}$.

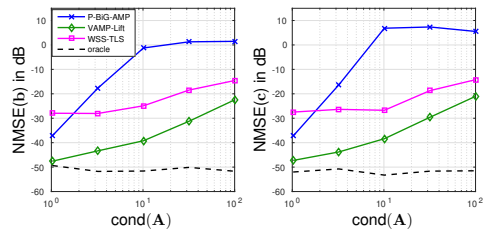

(b) NMSE vs. cond($\mathbf{A}$) at $M/P = 0.6$.

Figure 4: Compressive sensing with matrix uncertainty

estimation of $\mathbf{c}$ under oracle knowledge of support($\mathbf{c}$) and $\mathbf{b}$. For $b_1 = \sqrt{20}$, $L = 11$, $P = 256$, $K = 10$, i.i.d. $\mathcal{N}(0,1)$ matrix $\mathbf{A}$, and SNR = 40 dB, Fig. 4a shows the normalized MSE on $\mathbf{b}$ (i.e., $\mathsf{NMSE}(\mathbf{b}) := \mathbb{E}\|\widehat{\mathbf{b}} - \mathbf{b}^0\|^2 / \mathbb{E}\|\mathbf{b}^0\|^2$) and $\mathbf{c}$ versus sampling ratio $M/P$. This figure demonstrates that lifted VAMP and PBiGAMP perform close to the oracles and much better than WSS-TLS.

Although lifted VAMP performs similarly to PBiGAMP in Fig. 4a, its advantage over PBiGAMP becomes apparent with non-i.i.d. $\mathbf{A}$. For illustration, we repeated the previous experiment, but with $\mathbf{A}$ constructed using the SVD $\mathbf{A} = \mathbf{U}\mathrm{Diag}(\mathbf{s})\mathbf{V}^{\mathsf{T}}$ with Haar distributed $\mathbf{U}$ and $\mathbf{V}$ and geometrically spaced $\mathbf{s}$. Also, to make the problem more difficult, we set $b_1 = 1$. Figure 4b shows the normalized MSE on $\mathbf{b}$ and $\mathbf{c}$ versus cond($\mathbf{A}$) at $M/P = 0.6$. There it can be seen that lifted VAMP is much more robust than PBiGAMP to the conditioning of $\mathbf{A}$.

We next consider the *self-calibration* problem [39], where the measurements take the form

$$\mathbf{y} = \mathrm{Diag}(\mathbf{Hb})\mathbf{\Psi}\mathbf{c} + \mathbf{w} \in \mathbb{R}^M. \tag{24}$$

Here the matrices $\mathbf{H} \in \mathbb{R}^{M \times L}$ and $\mathbf{\Psi} \in \mathbb{R}^{M \times P}$ are known and the objective is to recover the unknown vectors $\mathbf{b}$ and $\mathbf{c}$. Physically, the vector $\mathbf{Hb}$ represents unknown calibration gains that lie in a known subspace, specified by $\mathbf{H}$. Note that (24) is an instance of (22) with $\mathbf{\Phi}_l = \mathrm{Diag}(\mathbf{h}_l)\mathbf{\Psi}$, where $\mathbf{h}_l$ denotes the $l$th column of $\mathbf{H}$. Different from "CS with matrix uncertainty," all elements in $\mathbf{b}$ are now unknown, and so WSS-TLS [42] cannot be applied. Instead, we compare lifted VAMP to the SparseLift approach from [39], which is based on convex relaxation and has provable guarantees. For our experiment, we generated $\mathbf{\Psi}$ and $\mathbf{b} \in \mathbb{R}^L$ as i.i.d. $\mathcal{N}(0,1)$; $\mathbf{c}$ as $K$-sparse with $\mathcal{N}(0,1)$ nonzero entries; $\mathbf{H}$ as randomly chosen columns of a Hadamard matrix; and $\mathbf{w} = \mathbf{0}$. Figure 3 plots the success rate versus $L$ and $K$, where "success" is defined as $\mathbb{E}\|\widehat{\mathbf{c}}\widehat{\mathbf{b}}^{\mathsf{T}} - \mathbf{c}^0(\mathbf{b}^0)^{\mathsf{T}}\|_F^2 / \mathbb{E}\|\mathbf{c}^0(\mathbf{b}^0)^{\mathsf{T}}\|_F^2 < -60$ dB. The figure shows that, relative to SparseLift, lifted VAMP gives successful recoveries for a wider range of $L$ and $K$.

## 6  Conclusions

We have extended the analysis of the method in [24] to a class of non-separable denoisers. The method provides a computational efficient method for reconstruction where structural information and constraints on the unknown vector can be incorporated in a modular manner. Importantly, the method admits a rigorous analysis that can provide precise predictions on the performance in high-dimensional random settings.

## Acknowledgments

A. K. Fletcher and P. Pandit were supported in part by the National Science Foundation under Grants 1738285 and 1738286 and the Office of Naval Research under Grant N00014-15-1-2677. S. Rangan was supported in part by the National Science Foundation under Grants 1116589, 1302336, and 1547332, and the industrial affiliates of NYU WIRELESS. The work of P. Schniter was supported in part by the National Science Foundation under Grant CCF-1527162.

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
