[Reviews · NeurIPS 2018]

Reviewer 1



Update: After reading the rebuttal of the authors, I think the authors have addressed my comments satisfactorily. I've increased my score of the paper to 7. ---------------------------------- Original comments: Summary: This paper provides a proof for the state evolution of plug-in denosing-based VAMP algorithm. The results extend the analyses in a previous paper by a subset of the authors. It is well written and the numerical results look nice. Here are my detailed comments: 1. Is it fair to say that the main difference between the current paper and [21] is related to the “convergent under Gaussian noise” assumption (in Definition 2), and other parts of the proof are parallel to those of [21]? 2. The main contribution of the paper is a proof of state evolution under certain conditions. The authors then provide simulations to show the performance advantage of the proposed algorithm over other algorithms. However, in my view, there seem to be some disconnections between theory and simulations: (1) In the analysis, it is assumed that the denoiser is Lipschitz continuous. However, the denoisers under study (such as the BM3D denoiser and the rank one AMP denoiser for bilinear estimation) doesn’t seem to satisfy such continuity assumptions. The possible discontinuity can have an impact on how you compute the derivatives required by the algorithm. This issue is discussed in [10] and a Monte Carlo method was proposed to address the discontinuity. I’m wondering what’s the impact of discontinuity on the performance of the proposed algorithm? Did you use similar ideas? (2) It is assumed that $A$ is right rotationally invariant. However, the matrixes involved in the numerical results (particularly the one in the bilinear estimation example) are quite structured. How does such structure affect the accuracy of the state evolution analysis? (3) The main contribution of the paper is a state evolution theory under certain assumptions. However, the simulations are carried out under conditions (seem (1) (2) above) different from the analysis. It’s not clear whether the theory can still provide good prediction of the empirical performance. To make the story complete, instead of comparing the empirical performance of the algorithm relative to others, I think it’s better to add numerical results that verify the SE predictions. 3. In theorem 1, it is assumed that $\bar{\alpha_{1k}}\in(0,1)$. This assumption seems quite strong. How can one check whether it holds or not? For example, does it hold for the denoisers involved in the numerical results of this paper? 4. The assumption in (13) seem to be extremely strong: it assumes that the initial estimate is an observation of the true signal correction by iid Gaussian noise. Is this practical? 5. The authors cite both [18] and [21]. Sometimes the authors refer to [18] and sometimes to [21]. But it seems that [18] is basically a short version of [21]. It is necessary to include both references?

Reviewer 2



AMP is an algorithm for linear estimation that is very powerful under certain assumptions of the matrix. It is being used both in some practical problems, and for theoretical understanding of high-dimensional statistical models. With respect to AMP, the VAMP algorithm works for a larger class of matrices and it thus particularly promising. One very important property of both AMP and VAMP is the existence of the so called state evolution which is a deterministic set of equations tracking the error in a given iteration of the algorithm. Both AMP and VAMP are able to take into account various regularisations and prior information on the unknown signal. In the simplest and most restrictive form the prior/regularizers needed to be separable for the state evolution to hold. This constraint was relaxed in previous works for the case of the AMP algorithm. The present work relaxed the separability constraint for VAMP and proves the state evolution for a class of non-separable denoisers. This is a solid work that inscribes in a line of work on the AMP and VAMP type of algorithms. The paper has several caveats that the authors should explain or amend: ** The paper has a theory part where the state evolution is proven (for space purposed most of the proofs are in the supplement, which is no problem), and a theory part that shows performance of the algorithm for 3 typical problems that can be treated with this type of algorithms. THe experimental part has comparison to other state-of-the-art algorithms. Since the main point of the paper is the state evolution, why doesn't the experimental part compare to the result of the state evolution? ** The sentence "whenever the SE has a unique fixed point, the estimates generated by AMP converge to the Bayes optimal estimate of x0 from y." does nto seem correct. First, there are situations when AMP runs with a denoiser that does not correspond to a probability distribution that generated the signal, that even if there is a unique fixed point the resulting error is not as good as the Bayes optimal one. Second, which theorem in refs. [6,7] states that unique fixed point of SE implies optimality and under what conditions? ** The authors say: "Unfortunately, the SE analyses of VAMP in [23] and its extension in [24] are limited to separable denoisers." Is seems to me that the work of Rangan, Fletcher'????? can be seen a applying VAMP with a non-separable prior. Why is this not considered by the authors? Details: Page 2: "of of [18]" repetition. In the conclusion it would be useful to remind that there are strict conditions on A and the denoiser under which the analysis of the paper holds. Otherwise a sentence as "the method admits a rigorous analysis that can provide precise predictions on the performance in highdimensional random settings." may be misleading. ------ My concerns and questions were well answered in the rebuttal, I increased my overall score from 6 to 7.

Reviewer 3



This work proves that the distribution of the intermediate variables within Vector Approximate Message Passing (VAMP) can be predicted by a state-evolution framework, even with non-seperable denoisers. This task is accomplished by showing that a variety of interesting denoisers are uniformly Lipschitz continuous (Def 1) and convergent under gaussian noise (Def 2). Under these conditions, the authors show that the covariance of the terms p and q defined in equation (18) can be predicted using equations (19a-d) (p is known as the effective error elsewhere in the AMP literature). In effect, this lets one predict the per iteration MSE of the algorithm. While the numerical results presented in this paper are interesting, they seem largely irrelevant to the rest of the paper. The results show that (D)-VAMP offers the same performance as (D)-AMP with Gaussian measurement matrices but can also handle ill-conditioned measurement matrices. However, this only demonstrates that (D)-AMP and (D)-VAMP have the same fixed points; it doesn't say anything about (D)-VAMP's state evolution. Moreover, the results make no mention of the state evolution whatsoever. These experiments need to be rethought (or at least better tied to the rest of the paper). Overall, this is a potentially important paper that is held back by its experimental validation. Understanding (D)-VAMP theoretically is important and timely; (D)-VAMP may enable the application of the powerful (D)-AMP algorithm to real-world problems. Unfortunately, the experimental sections only show that (D)-VAMP works, which has been established elsewhere, e.g., "Denoising-based Vector AMP". The paper is well written, some minor issues: The first time the divergence is defined I believe it should be "normalized divergence" I'm not sure [30] is the intended denoising CNN reference. Line 120 dropped a preposition